# Impact of COVID-19 Pandemic on Children’s Fundamental Motor Skills: A Study for the Taiwanese Preschoolers Teachers

**DOI:** 10.3390/ijerph20186764

**Published:** 2023-09-15

**Authors:** Shu-Yu Cheng, Hsia-Ling Tai, Tsung-Teng Wang

**Affiliations:** 1Graduate Institute of Sport Training, University of Taipei, Taipei 111036, Taiwan; shu-yu@utaipei.edu.tw; 2Graduate Institute of Physical Education, University of Taipei, Taipei 100234, Taiwan; danatai1008@utaipei.edu.tw

**Keywords:** physical movement courses, physical activity, pandemic, stability, locomotor, manipulative

## Abstract

The outbreak of the COVID-19 pandemic has resulted in reduced opportunities for children to engage in fundamental motor skills [FMS]. This prolonged inactivity and restriction of play can have serious consequences for children’s physical and mental health. The purpose of this study was to explore teaching strategies during the pandemic, whether there were differences in children’s motor development, and the differences in the implementation of physical movement courses before and during the pandemic from the perspective of preschool teachers. This study was a retrospective study using an internet survey, and participants comprised 2337 preschool teachers. The statistical methodology of this study included descriptive statistics, the dependent *t*-test, and the independent *t*-test. The results showed that regardless of the time, frequency, activity intensity, and frequency of outdoor courses, the results from before the pandemic was better than those taken during the pandemic. Only the “frequency of implementing physical movement courses indoors every week” had not been affected by the pandemic. This study also obtained the performance of “children’s fitness”, “overall performance of physical movement ability”, “stability movement skills”, “locomotor movement skills”, and “manipulative movement skills”. All were better before the pandemic than during the pandemic. During the COVID-19 pandemic, mixed-age classes performed better than same-age classes in terms of frequency, time, intensity, outdoor course implementation, and physical fitness. Public schools performed better than private schools in terms of frequency, time, intensity, outdoor course implementation, and fundamental motor skills performance. Private schools implemented physical movement courses indoors every week, which was more than public schools. Excepting the frequency of implementing physical movement courses indoors every week, fewer than schools with five classes performed better than those who had more than schools with six classes. Finally, rural schools were better than urban schools in the implementation of outdoor courses and fundamental motor skills performance. Therefore, we suggest that in response to the pandemic, teachers should further improve their professionalism and use diversified teaching methods, and guide students to be willing to learn and improve their skill performance.

## 1. Introduction

It should be children’s right to play freely, but the events of the COVID-19 pandemic deprived children of the right to play. The Central Epidemic Control Center (CECC) confirmed the first case of COVID-19 in Taiwan on 21 January 2020 [1]. The World Health Organization (WHO) [2] was declared a global public health emergency on 11 March 2020. The cumulative number of confirmed cases in Taiwan was 10,241,503, with 8390 deaths (from 21 January 2020 to 3 July 2023) [3]. The pandemic has had widespread psychological, health, social, and economic impacts [4,5,6]. The World Organization for Early Childhood Education [OMEP] report states that the pandemic has affected and changed the lives and routines for young children, often with huge limitations on children’s vital need to play, move, and relate with their peers, and with reduced social contacts beyond the family [7]. This initiated the beginning of a long period of inactivity and play restriction, which can lead to serious consequences for children’s physical and mental health [8]. This could possibly cause anxiety, depression, stress, disturbances in sleep and appetite, interpersonal and environmental restraint, sensorial deprivation, and neglect [9,10,11,12,13].

In order to prevent the spread of COVID-19 during the pandemic, the Taiwanese government implemented several isolation control methods including limiting mass gatherings and outdoor activities, closing schools, maintaining physical distance, and even lockdowns. Children across all levels of schools and public and private preschools across the country were stopped from attending classes from 19 May 19 to 28 May 2021 [14], which has had a major impact on the way preschool students attend classes. Isolation measures have greatly reduced the space for children’s activities, and the implementation of physical activity courses has been hindered. During the isolation period, children are deprived of the opportunity to play games with their peers, and their right to be taught physical movement courses is limited. The World Health Organization [WHO] has developed physical activity guidelines for preschool-aged children (3–5 years) that suggests at least 180 min in a variety of types of physical activity, 60 min of which should be devoted to moderate-to-vigorous-intensity physical activity [15]. During the COVID-19 quarantine, relevant studies have confirmed the impact of the pandemic on children, such as reduced physical activity, increased sedentary time, screen time, and sleep time [16,17,18,19,20].

The global burden of sequelae of physical inactivity is enormous. A number of studies have pointed out that physical diseases caused by lack of physical activity include chronic diseases such as cardiovascular disease and metabolic syndrome [21,22,23]. Therefore, the reduction in physical activity caused by related regulatory measures will also affect the deterioration of health behaviors. In children, higher levels of physical activity are positively associated with healthy quality of life, whereas sedentary behavior negatively affects healthy quality of life [24]. It shows that the higher the frequency of physical activity or the less sedentary time, the better the quality of life.

For educators, the pandemic also had a great impact on their teaching in physical movement. A study of teachers, parents, and principals found that the risk of danger in the traditional classroom plus bad weather were considered barriers by educators [25]. Promoting physical activity education was challenged due to equipment, space, and curriculum enrichment opportunities losses [26]. In addition, the implementation of pandemic prevention measures in the school environment may delay the progress of the pandemic. However, this may also result in students spending less time in school, making teaching more difficult [27].

Nevertheless, the relevant research on preschool during the pandemic is still quite scarce, especially the implementation difficulties of physical movement courses and the differences in children’s motor skills before and during the pandemic. This study focuses on the impact of the pandemic on children’s fundamental motor skills and other physical activities, and whether there will be a gap in performance before and during the pandemic. Therefore, by investigating the current situation of physical activity teaching in preschool, we will use the observations of preschool teachers as a benchmark to explore the comprehensive impact of the pandemic on children’s physical performance in preschool. At the same time, we also hope to explore whether the arrangement of physical movement courses in preschool is also affected by the pandemic. Thus, this study will comprehensively compare the differences in the comprehensive performance of children’s physical movement development before and during the pandemic, and expect to understand various points of guidance from preschools in the face of the pandemic, as well as how the teaching of children’s physical movement courses strengthened the program.

## 2. Materials and Methods

### 2.1. Sociodemographic Variables

This was a retrospective study regarding how children’s fundamental motor skills and physical movement courses taught by Taiwanese preschool teachers changed during the COVID-19 pandemic. The participants of this study were required to have more than three years of teaching experience in preschool. The main teaching students were children aged 2–6.

A sample of 2673 preschool teachers took part in the study with an average teaching experience of 15 years from different parts of Taiwan. A total of 2337 valid questionnaires were received, excluding those with incomplete answers, and the recovery rate of valid samples was 87.4%. An online survey was created and distributed using convenience sampling from 7 November to 10 December 2022 via the Surveycake platform. In order to increase the return rate of the questionnaire, social media (Facebook, Line, Messenger) and email were used to invite educational institutions. Before completing the survey, all study participants were given information about the link to read more details about the study and accepted the invitation online to participate in the study anonymously. The study was conducted according to the provisions of the Research Ethic Committee of the University of Taipei expressed its approval (Decision No.: IRB-2022-093).

### 2.2. Questionnaire

The research was based on the self-compiled questionnaire “2–6 years old children’s fundamental motor skills and curriculum questionnaire”. The questionnaire was assessed using a 5-point Likert scale in order to understand the impact of the pandemic on children.

Each participant subjectively completed the questionnaire on basic information, the teaching of physical movement courses, the performance of fundamental motor skills, how these were implemented during the pandemic (from January 2020 to November 2022), and using the same questions asked retrospectively, remembering the implementation before the pandemic (before January 2020).

In the design of the questionnaire, each participant was asked to answer the same questions on the performance of the children before and during the pandemic.

Each participant completed the questionnaire in approximately 10 min. The questionnaire comprised 3 sections (14 questions):
1.Sociodemographic characteristics: There were four questions in this section.

Included class types, school types, school size, and school location.
2.Curriculum implementation: There were five questions in this section. Each question was assessed using a Likert scale (1 to 5). The implementation of physical movement courses before and during the pandemic was compared using the same questions raised retrospectively. Points included time to implement physical movement courses (under 20, 20–29, 30–39, 40–49 min more than 50 min), frequency of physical movement courses per week (1–5 days), intensity of activity for weekly physical movement courses (barely there light, moderate, hard and very hard), frequency of implementing physical movement courses outdoors every week (1–5 days), and frequency of implementing physical movement courses indoors every week (1–5 days).3.Fundamental motor skills: There were five questions in this section. Each question was assessed using a Likert scale (poor, fair, good, very good, excellent). The performance of children’s fundamental motor skills was compared before and during the pandemic using the same questions raised retrospectively. Points included children’s fitness; children’s overall performance in physical movement ability; children are performing stability movement skills such as bending, squatting, balancing; children performing locomotor movement skills such as walking, running, jumping, climbing; children performing manipulative movement skills such as catching, throwing, kicking, striking.

### 2.3. Statistical Analysis

Analysis participants with complete data for all sections were included in the analysis. Descriptive data were expressed as mean (standard deviation, SD) for continuous variables, and categorical variables were presented as frequency and percentage. Dependent sample *t*-test was used to evaluate significant changes in the implementation of physical movement courses and fundamental motor skills before the pandemic (before January 2020) vs. during the pandemic (from January 2020 to November/December 2022). An independent sample *t*-test was utilized to compare the difference in implementing physical movement courses and the performance of fundamental motor skills during the pandemic with different background variables. Statistical significance was set at *p* < 0.05. All analyses were performed using IBM SPSS Statistics for Windows version 25.0 (IBM Corp., Armonk, New York, NY, USA).

## 3. Results

As can be seen from Table 1, the dependent sample t-test was utilized to compare the time difference in implementing physical movement courses before and during the pandemic. The results showed that the results from “before the pandemic” were better than those from “during the pandemic” (*t* = 9.14, * *p* < 0.001). The results for “Frequency of physical movement courses per week” also showed that “before the pandemic” performed more favorably than “during the pandemic” (*t* = 8.61, * *p* < 0.001); moreover, for “Frequency of implementing physical movement courses outdoors every week”, the results from “before the pandemic” were also better than those “during the pandemic” (*t* = 16.81, * *p* < 0.001). The results for “Intensity of activity for weekly physical movement courses” were also the same, which “before the pandemic” were better than those for “during the pandemic” (*t* = 21.92, * *p* < 0.001). Only the “Frequency of implementing physical movement classes indoors every week” had no significant difference; it was found that the frequency of carrying out physical movement courses “indoors” before and during the pandemic was similar, and was not affected by the pandemic.

As can be seen from Table 2, Comparing the differences in the performance of children’s fundamental motor skills before and during the pandemic, the dependent sample *t*-test was carried out to find out the difference in fundamental motor skills performance. The results showed that responses taken “before the pandemic” were better than those taken “during the pandemic” (*t* = 45.86, * *p* < 0.001). This showed that the physical performance of children before the pandemic was better than during the pandemic. It is inferred that the reason was that the control measures during the pandemic restricted children’s activities and affected their physical fitness. Additionally, in the results for “Overall performance of physical movement ability”, responses taken from “before the pandemic” were better than those taken “during the pandemic” (*t* = 44.64, * *p* < 0.001); thus, not only was physical fitness affected, but the overall performance of physical movement ability was also negatively affected, and further subdivided the fundamental motor skills into “stability movement skills”, “locomotor skills”, and “manipulative movement skills” for further discussion. For “stability movement skills”, results taken from “before the pandemic” were better than those taken “during the pandemic” (*t* = 40.77, * *p* < 0.001). The results for “locomotor motor skills” were also the same, namely that responses taken “before the pandemic” were better than those taken “during the pandemic” (*t* = 42.92, * *p* < 0.001). The same results were obtained for “manipulative movement skills”, showing that results taken “before the pandemic” were better than those taken “during the pandemic” (*t* = 40.72, * *p* < 0.001). The performance before the pandemic was better than during the pandemic, because the pandemic control measures restricted the activities of children and restricted the development of their motor skills.

Figure 1 illustrates changes in the implementation of physical movement courses and the performance of fundamental motor skills before and during the pandemic. As shown in the figure, except for the fact that there is no difference in the frequency of implementing physical movement courses indoors every week, the performance of all other items during the pandemic was worse than before the pandemic and reached a significant level of 0.001.

We carried out a comparison of differences in the implementation of physical movement courses during the COVID-19 pandemic considering different background variables. The independent sample *t*-test was used to find out the difference in implementation of physical movement courses. The results were as follows. In addition to the frequency of implementing physical movement courses indoors every week, mixed-age classes performed better than same-age classes. Regardless of the frequency, time, intensity, or outdoor course implementation, public schools performed better than private schools. Private schools implemented physical movement courses indoors every week, more frequently than public schools. Excepting the frequency of implementing physical movement courses indoors, fewer than schools with five classes performed better than those with more than schools with six classes. Finally, concerning the frequency of implementing physical movement courses indoors every week, urban schools were better than rural schools. Rural schools were better than urban schools in frequency of implementing physical movement courses outdoors every week. All information is presented in Table 3 and Figure 2.

We compared the differences in performance of children’s fundamental motor skills during the pandemic, considering different background variables. The independent sample *t*-test was used to find out the difference in performance of children’s fundamental motor skills. Mixed-age classes performed better than same-age classes in physical fitness. There is no difference in school type between items. Schools with fewer than five classes performed better than schools with more than six classes in terms of physical fitness, physical movement ability, stability movement skills, locomotor skills, and manipulative movement skills. In addition, rural schools performed better than urban schools in terms of physical fitness, physical movement ability, stability movement skills, locomotor skills, and manipulative movement skills. See Table 4 and Figure 3 for detailed data.

## 4. Discussion

The differences in the implementation of physical movement courses before the pandemic, regardless of the courses time, frequency, activity intensity, and frequency of outdoor courses, were higher than those during the pandemic. Only the “Frequency of implementing physical movement courses indoors every week” had not been affected by the pandemic. Because of the quarantine policy, children were unable to play and explore outdoors. Therefore, the mode of activity switched from “outdoor play mode” to “indoor play mode” [4]. Activities that are high-intensity or require a high level of exertion (such as full competition) present a higher level of risk of catching and spreading COVID-19 than lower-intensity activities [28]. Teachers may believe that children lack the ability to protect themselves from the pandemic, meaning that some teachers do not want children to have too many opportunities to be in close proximity with each other. Therefore, during the pandemic, the time and frequency of course implementation were reduced to reduce the risk of infection, but this also deprived children of opportunities for activities. Preschool students spent only 4% of the day in indoor active play opportunities, and 8% in outdoor active play opportunities [29]. Therefore, how to give children more space and opportunities for activities while maintaining an appropriate distance is a topic that preschool teachers must face.

The effectiveness of preschool work depends largely on the education, skills, and personal inclinations of teachers. The situation caused by the pandemic has confirmed how necessary it is for teachers to continue to develop their professional skills, to continue learning throughout their lives, to see and follow changes, to improve the quality of teaching, and to look for new forms and methods of teaching [30].

This study also obtained the performance of “children’s fitness”, “overall performance of physical movement ability”, “stability motor skills”, “locomotor motor skills”, and “manipulative motor skills”. All were better before the pandemic than during the pandemic. The COVID-19 pandemic has clearly affected children’s lives. There are many studies on the reduced physical activity of children during the COVID-19 pandemic [31,32,33,34].

A study in Japan (ages 3 to 5) found that all grades scored higher before than during the pandemic. These findings suggest that the COVID-19 pandemic impeded the development of fundamental motor skills, especially manipulative skills [35]. In addition, research on Chilean elementary school children also found that locomotor motor skills before the pandemic was better than during the pandemic [36]. Research on Portuguese children also found that during the pandemic, the locomotor motor skills for young children and both the locomotor and manipulative motor skills for the more motor-proficient children were worse than before the pandemic [37]. Research on Austrian elementary school children also found that their performance during the epidemic was worse than before the epidemic, regardless of their performance in countermovement jumps, agility runs, 6 min runs, stand-and-reach tests, or throw and catch [38]. For most people, many movements do not develop “automatically” or because of biology. Improvement through maturity requires practice opportunities, encouragement, guidance, a learning environment, and supportive teachers. It is necessary to recognize and be able to analyze the need for movement and to give appropriate guidance [39]. Research found that supporting higher physical activity levels and providing quality learning experiences to improve FMS are vital for school-aged children (5–7 years) [40]. Therefore, the long-term goal of teacher curriculum design should be to guide children to develop habitual physical activity. Ensuring children can access outdoor spaces and maintaining the mental health of their parents gives preschoolers the opportunity to engage in healthy physical activity that meets global standards [41].

This study also found that mixed-age classes performed better than same-age classes, n no matter the frequency, time, intensity, outdoor of the course implementation, and level of physical fitness. A study on Chinese children reported that the mixed-age classrooms had significantly more positive interactions and more control than the same-age classrooms [42]. From the perspective of motor development, children in lower grades can easily observe and imitate the exercise methods of children in higher grades [43]. This may be the reason why the mixed-age classes performed better than the same-age classes in physical movement courses. Public schools performed better than private schools regardless of time, frequency, intensity, and implementation of outdoor courses. Private schools implemented physical movement courses indoors every week, which was more than public schools. Due to the regulations of the Taiwan Ministry of Education, preschools must have 30 min of high-intensity gross muscle activity. The implementation of curricula in public schools can effectively implement policies, thus achieving better results.

In different school sizes, this study found that in addition to the frequency of implementing physical movement courses indoors every week, schools with fewer than five classes were better than those with more than six classes. Schools with fewer than five classes had larger spaces and fields for activities that led to such results. According to the research, preschool environmental factors significantly affect children’s physical activity and levels of physical fitness. This survey conducted with 4287 participants found that children with large playgrounds in these kindergartens had better levels of physical fitness [44]. Additional research on rural and urban areas found that as age increases, rural children are better performers in the standing long jump and tennis ball throw for distance than their urban peers [45].

Because of the impact of the pandemic, changes in school dynamics, and lifestyle, children experience more fear, stress, and serious consequences directly affecting their mental health [4]. According to this, the ecology of the school not only affects the physical and mental development of children, but also connects their future health, which has a huge impact. Moreover, under the conditions of limited contact during periods of pandemic control, children’s physical movement performance was not as good as before the pandemic. This study has repeatedly emphasized the importance of various environments for the development of children’s physical movements. It is hoped that parents and teachers will work hard to create opportunities for children. Providing an environment that is conducive to the development of fundamental motor skills can guide and stimulate children’s enthusiasm for physical activities, cultivate lifelong exercise, and benefit physical and mental health.

### Limitations

This study includes a large sample of teachers from every region in Taiwan. All teachers were contacted online; participant response bias may have affected the answers, due to these unusual events. In addition, this study did not use accelerometers or other measurement tools, and only measured children’s physical activities and behaviors based on teachers’ subjective standards, which may affect the results of the study. This study uses a retrospective research method, meaning that there may also be a risk of recall bias.

## 5. Conclusions

This study is one of the very few studies from the perspective of preschool teachers exploring the control measures during the pandemic, whether there are differences in the development of children’s movements, and the differences in the implementation of physical movement courses before and during the pandemic. The results of this study showed that after the implementation of control measures during the COVID-19 pandemic, the development of children’s physical movements before the pandemic was better than during the pandemic, and the intensity and frequency of physical movement courses taught by preschool teachers before the pandemic were better than during the pandemic.

Because the teacher’s attitude and guidance are very important in the development of children’s movements, many movement skills do not develop automatically with age, and require teachers’ guidance and more practice to become proficient. The frequency, time and intensity of children’s activities will affect the child’s basic movements. Development matters. Regardless of the changes caused by the educational environment or the needs of pandemic prevention, preschool teachers should enhance their professional and diverse teaching experience, adapt to changes, and guide students to develop lifelong exercise habits and motor skills as the long-term teaching goal.

Therefore, we suggest that in the future, we should develop teaching materials and courses, design reference materials for teachers, improve the quality of courses in situations such as the pandemic control period, and enable teachers to have the ability to fully grasp the courses on physical movements. These findings can support future pandemics of infectious diseases. The teaching mode and professional growth needs of teachers’ physical exercise courses have been resolved, and it is the key demand in order to promote children’s healthy exercise behavior and movement development.

## Figures and Tables

**Figure 1 ijerph-20-06764-f001:**
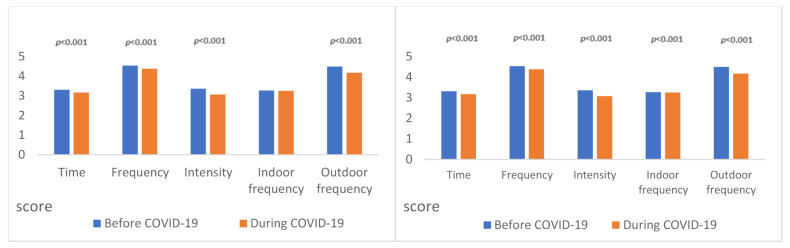
Change in implementation of physical movement courses and performance of fundamental motor skills (pre vs. during pandemic).

**Figure 2 ijerph-20-06764-f002:**
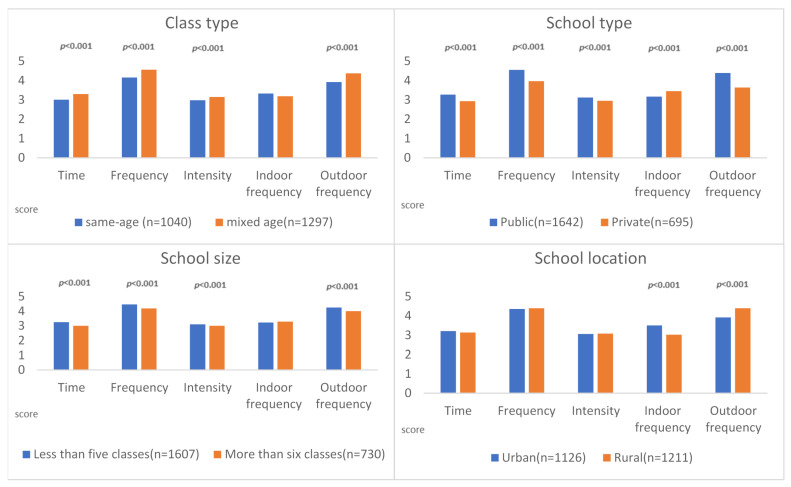
Change in implementation of physical movement courses with different background variables (during pandemic).

**Figure 3 ijerph-20-06764-f003:**
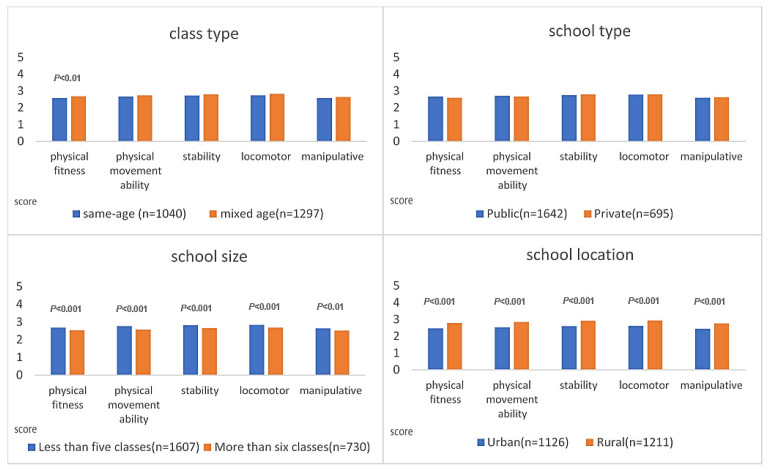
Change in fundamental motor skills performance with different background variables (during pandemic).

**Table 1 ijerph-20-06764-t001:** Comparison of differences in implementation of physical movement courses before and during the COVID-19 pandemic.

Physical Movement Courses	Before COVID-19 (*n* = 2337)		During COVID-19 (*n* = 2337)		*t*	*p*-Value
	M	SD	M	SD		
Time	3.31	0.88	3.17	0.96	9.14 ***	<0.001
Frequency	4.53	1.04	4.38	1.14	8.61 ***	<0.001
Intensity	3.36	0.57	3.07	0.61	21.92 ***	<0.001
Indoor frequency	3.27	1.37	3.25	1.39	0.83	0.41
Outdoor frequency	4.49	0.94	4.17	1.21	16.81 ***	<0.001

*** *p* < 0.001.

**Table 2 ijerph-20-06764-t002:** Comparing the differences in children’s fundamental motor skills performance before and during the pandemic.

Motor Skill Performance	Before COVID-19 (*n* = 2337)		During COVID-19 (*n* = 2337)		*t*	*p*-Value
	M	SD	M	SD		
Physical fitness	3.63	0.88	2.64	0.99	45.86 ***	<0.001
Physical movement ability	3.64	0.85	2.70	0.97	44.64 ***	<0.001
Stability	3.61	0.83	2.77	0.97	40.77 ***	<0.001
Locomotor	3.71	0.85	2.79	1.00	42.92 ***	<0.001
Manipulative	3.44	0.85	2.61	0.98	40.72 ***	<0.001

*** *p* < 0.001.

**Table 3 ijerph-20-06764-t003:** Comparison of differences in implementation of physical movement courses during the COVID-19 pandemic with different background variables.

	Time		Frequency		Intensity		Indoor Frequency		Outdoor Frequency	
Characteristic	During COVID-19 Mean (SD)	*p*	During COVID-19 Mean (SD)	*p*	During COVID-19 Mean (SD)	*p*	During COVID-19 Mean (SD)	*p*	During COVID-19 Mean (SD)	*p*
Class type:										
Same age (n = 1040)	3.01 (0.98)	<0.001 ***	4.16 (1.26)	<0.001 ***	2.98 (0.61)	<0.001 ***	3.33 (1.38)	0.012	3.92 (1.32)	<0.001 ***
Mixed age (n = 1297)	3.30 (0.93)		4.56 (0.99)		3.15 (0.59)		3.19 (1.39)		4.37 (1.07)	
School type:										
Public (n = 1642)	3.27 (0.92)	<0.001 ***	4.55 (1.00)	<0.001 ***	3.12 (0.59)	<0.001 ***	3.17 (1.38)	<0.001 ***	4.39 (1.01)	<0.001 ***
Private (n = 695)	2.93 (1.03)		3.97 (1.32)		2.95 (0.62)		3.45 (1.38)		3.64 (1.46)	
School size:										
Fewer than five classes (n = 1607)	3.25 (0.94)	<0.001 ***	4.47 (1.09)	<0.001 ***	3.10 (0.59)	<0.001 ***	3.23 (1.40)	0.319	4.25 (1.19)	<0.001 ***
More than six classes (n = 730)	3.00 (0.98)		4.19 (1.22)		3.00 (0.63)		3.29 (1.36)		4.00 (1.22)	
School location:										
Urban (n = 1126)	3.21 (0.97)	0.59	4.36 (1.13)	0.40	3.06 (0.59)	0.52	3.50 (1.33)	<0.001 ***	3.92 (1.34)	<0.001 ***
Rural (n = 1211)	3.14 (0.96)		4.40 (1.14)		3.08 (0.62)		3.02 (1.40)		4.40 (1.01)	

*** *p* < 0.001.

**Table 4 ijerph-20-06764-t004:** Comparing the differences in children’s fundamental motor skills performance during the pandemic with different background variables.

	Physical Fitness		Physical Movement Ability		Stability Movement Skills		Locomotor Movement Skills		Manipulative Movement Skills	
Characteristic	During COVID-19 Mean (SD)	*p*	During COVID-19 Mean (SD)	*p*	During COVID-19 Mean (SD)	*p*	During COVID-19 Mean (SD)	*p*	During COVID-19 Mean (SD)	*p*
Class type:										
Same age (n = 1040)	2.58 (0.97)	<0.007 **	2.66 (0.97)	0.041	2.73 (0.99)	0.063	2.75 (1.02)	0.055	2.58 (0.99)	0.139
Mixed age (n = 1297)	2.69 (0.99)		2.74 (0.97)		2.81 (0.96)		2.83 (0.99)		2.64 (0.96)	
School type:										
Public (n = 1642)	2.66 (1.01)	0.233	2.71 (0.98)	0.566	2.76 (0.97)	0.521	2.79 (1.00)	0.355	2.60 (0.97)	0.758
Private (n = 695)	2.60 (0.95)		2.68 (0.96)		2.80 (0.99)		2.80 (1.02)		2.63 (1.00)	
School size:										
Fewer than five classes (n = 1607)	2.69 (1.00)	<0.001 ***	2.76 (0.97)	<0.001 ***	2.82 (0.96)	<0.001 ***	2.84 (1.00)	<0.001 ***	2.65 (0.97)	<0.003 **
More than six classes (n = 730)	2.54 (0.96)		2.58 (0.96)		2.66 (0.99)		2.69 (0.99)		2.52 (0.97)	
School location:										
Urban (n = 1126)	2.48 (0.95)	<0.001 ***	2.54 (0.94)	<0.001 ***	2.60 (0.94)	<0.001 ***	2.62 (0.97)	<0.001 ***	2.44 (0.94)	<0.001 ***
Rural (n = 1211)	2.79 (1.00)		2.85 (0.97)		2.93 (0.98)		2.95 (1.01)		2.77 (0.98)	

*** *p* < 0.001 ** *p* < 0.01.

## Data Availability

Data are archived at the Graduate Institute of Physical Education, University of Taipei, Taiwan. If necessary, contact the author by e-mail tsungtengwang@yahoo.com.

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
