# Peer review of "Impact of COVID-19 Pandemic on Children’s Fundamental Motor Skills: A Study for the Taiwanese Preschoolers Teachers"

_ijerph, 2023, doi:10.3390/ijerph20186764_

Round 1

Reviewer 1 Report

It was evaluated the article titled “Impact of COVID-19 Pandemic on Children’s Fundamental Motor Skills: A Study for the Taiwanese Preschoolers Teachers”.

The proposal of this study was to explore the impact of teachers on children's Fundamental Motor Skills and physical activity performance during the pandemic, and to understand whether preschool teachers' physical activity teaching was affected by the pandemic. This study was a retrospective study using an internet survey, and 2337 preschool teachers were the participants.

The response to the proposal raised is obvious. A complete society had a large and negative impact due to COVID-19. My suggestion is to improve the proposal or rewrite it.

ABSTRACT: please, review the intro used.

INTRO: please, use new and more articles talking about the problems caused by COVID.

M&M

There are results presented in this section. Review them.

Was the questionnaire used previously validated?

RESULTS - well done

DISCUSSION: include more articles showing the problems found in the society, such as:

1. COVID-19 pandemic impact on children and adolescents' mental health: Biological, environmental, and social factors 

Prog Neuropsychopharmacol Biol Psychiatry. 2021 Mar 2;106:110171. doi: 10.1016/j.pnpbp.2020.110171

2. Impact of COVID-19 on Portuguese Dental Students: A Cohort Study. 2023 Healthcare 11(6):818 - DOI: 10.3390/healthcare11060818

CONCLUSION: ok!

Author Response

Thank you for the careful reading and constructive suggestions, the following are the revisions and responses to the relevant questions.

1.The abstract and introduction had been rewritten, in addition to adding more relevant literature, and supplementing the impact of the pandemic on children.

2.References and additions to valuable literature provided by reviewers.

3.Add the following articles.

  1. de Figueiredo, C. S., Sandre, P. C., Portugal, L. C. L., Mázala-de-Oliveira, T., da Silva Chagas, L., Raony, Í., Ferreira, E. S., Giestal-de-Araujo, E., Dos Santos, A. A., & Bomfim, P. O. (2021). COVID-19 pandemic impact on children and adolescents' mental health: Biological, environmental, and social factors. Progress in neuro-psychopharmacology & biological psychiatry, 106, 110171. https://doi.org/10.1016/j.pnpbp.2020.110171
  2. Hosseinzadeh, P., Zareipour, M., Baljani, E., & Moradali, M. R. (2022). Social Consequences of the COVID-19 Pandemic. A Systematic Review. Investigacion y educacion en enfermeria, 40(1), e10. https://doi.org/10.17533/udea.iee.v40n1e10
  3. Matsubayashi, T., Ishikawa, Y., & Ueda, M. (2022). Economic crisis and mental health during the COVID-19 pandemic in Japan. Journal of affective disorders, 306, 28–31. https://doi.org/10.1016/j.jad.2022.03.037
  4. Brooks, S. K., Webster, R. K., Smith, L. E., Woodland, L., Wessely, S., Greenberg, N., & Rubin, G. J. (2020). The psychological impact of quarantine and how to reduce it: rapid review of the evidence. Lancet (London, England), 395(10227), 912–920. https://doi.org/10.1016/S0140-6736(20)30460-8
  5. Samji, H., Wu, J., Ladak, A., Vossen, C., Stewart, E., Dove, N., Long, D., & Snell, G. (2022). Review: Mental health impacts of the COVID-19 pandemic on children and youth - a systematic review. Child and adolescent mental health, 27(2), 173–189. https://doi.org/10.1111/camh.12501
  6. Meherali, S., Punjani, N., Louie-Poon, S., Abdul Rahim, K., Das, J. K., Salam, R. A., & Lassi, Z. S. (2021). Mental Health of Children and Adolescents Amidst COVID-19 and Past Pandemics: A Rapid Systematic Review. International journal of environmental research and public health, 18(7), 3432. https://doi.org/10.3390/ijerph18073432
  7. Tang, S., Xiang, M., Cheung, T., & Xiang, Y. T. (2021). Mental health and its correlates among children and adolescents during COVID-19 school closure: The importance of parent-child discussion. Journal of affective disorders, 279, 353–360. https://doi.org/10.1016/j.jad.2020.10.016
  8. Li, B., Ng, K., Tong, X., Zhou, X., Ye, J., & Yu, J. J. (2023). Physical activity and mental health in children and youth during COVID-19: a systematic review and meta-analysis. Child and adolescent psychiatry and mental health, 17(1), 92. https://doi.org/10.1186/s13034-023-00629-4
  9. Park, A. H., Zhong, S., Yang, H., Jeong, J., & Lee, C. (2022). Impact of COVID-19 on physical activity: A rapid review. Journal of global health, 12, 05003. https://doi.org/10.7189/jogh.12.05003
  10. Wunsch, K., Kienberger, K., & Niessner, C. (2022). Changes in Physical Activity Patterns Due to the Covid-19 Pandemic: A Systematic Review and Meta-Analysis. International journal of environmental research and public health, 19(4), 2250. https://doi.org/10.3390/ijerph19042250
  11. Tulchin-Francis, K., Stevens, W., Jr, Gu, X., Zhang, T., Roberts, H., Keller, J., Dempsey, D., Borchard, J., Jeans, K., & VanPelt, J. (2021). The impact of the coronavirus disease 2019 pandemic on physical activity in U.S. children. Journal of sport and health science, 10(3), 323–332. https://doi.org/10.1016/j.jshs.2021.02.005
  12. DeBoer M. D. (2019). Assessing and Managing the Metabolic Syndrome in Children and Adolescents. Nutrients, 11(8), 1788. https://doi.org/10.3390/nu11081788
  13. Konuthula, D., Tan, M. M., & Burnet, D. L. (2023). Challenges and Opportunities in Diagnosis and Management of Cardiometabolic Risk in Adolescents. Current diabetes reports, 23(8), 185–193. https://doi.org/10.1007/s11892-023-01513-3
  14. Julian, V., Ciba, I., Olsson, R., Dahlbom, M., Furthner, D., Gomahr, J., Maruszczak, K., Morwald, K., Pixner, T., Schneider, A., Pereira, B., Duclos, M., Weghuber, D., Thivel, D., Bergsten, P., & Forslund, A. (2021). Association between Metabolic Syndrome Diagnosis and the Physical Activity-Sedentary Profile of Adolescents with Obesity: A Complementary Analysis of the Beta-JUDO Study. Nutrients, 14(1), 60. https://doi.org/10.3390/nu14010060
  15. Schwarzfischer, P., Gruszfeld, D., Stolarczyk, A., Ferre, N., Escribano, J., Rousseaux, D., Moretti, M., Mariani, B., Verduci, E., Koletzko, B., & Grote, V. (2019). Physical Activity and Sedentary Behavior From 6 to 11 Years. Pediatrics, 143(1), e20180994. https://doi.org/10.1542/peds.2018-0994
  16. Krishnaratne, S., Littlecott, H., Sell, K., Burns, J., Rabe, J. E., Stratil, J. M., Litwin, T., Kreutz, C., Coenen, M., Geffert, K., Boger, A. H., Movsisyan, A., Kratzer, S., Klinger, C., Wabnitz, K., Strahwald, B., Verboom, B., Rehfuess, E., Biallas, R. L., Jung-Sievers, C., … Pfadenhauer, L. M. (2022). Measures implemented in the school setting to contain the COVID-19 pandemic. The Cochrane database of systematic reviews, 1(1), CD015029. https://doi.org/10.1002/14651858.CD015029
  17. BrzÄ™k, A.; Strauss, M.; Sanchis-Gomar, F.; & Leischik, R. (2021). Physical Activity, Screen Time, Sedentary and Sleeping Habits of Polish Preschoolers during the COVID-19 Pandemic and WHO’s Recommendations: An Observational Cohort Study. International Journal of Environmental Research and Public Health, 18,
  18. Chen, X., Hong, X., Gao, W., Luo, S., Cai, J., Liu, G., & Huang, Y. (2022). Causal relationship between physical activity, leisure sedentary behaviors and COVID-19 risk: a Mendelian randomization study. Journal of translational medicine, 20(1), 216. https://doi.org/10.1186/s12967-022-03407-6
  19. Bozzola, E., Barni, S., Ficari, A., & Villani, A. (2023). Physical Activity in the COVID-19 Era and Its Impact on Adolescents' Well-Being. International journal of environmental research and public health, 20(4), 3275. https://doi.org/10.3390/ijerph20043275
  20. Teich, P., Fühner, T., Bähr, F., Puta, C., Granacher, U., & Kliegl, R. (2023). Covid Pandemic Effects on the Physical Fitness of Primary School Children: Results of the German EMOTIKON Project. Sports medicine - open, 9(1), 77. https://doi.org/10.1186/s40798-023-00624-1
  21. Carcamo-Oyarzun, J., Salvo-Garrido, S., & Estevan, I. (2023). Actual and Perceived Motor Competence in Chilean Schoolchildren before and after COVID-19 Lockdowns: A Cohort Comparison. Behavioral sciences (Basel, Switzerland), 13(4), 306. https://doi.org/10.3390/bs13040306
  22. Carballo-Fazanes, A., Rodrigues, L. P., Silva, R., Lopes, V. P., & Abelairas-Gómez, C. (2022). The Developmental Trajectory of Motor Competence of Children That Lived the COVID-19 Confinement Period: A Four-Year Follow-Up Study in Portuguese Children. Journal of functional morphology and kinesiology, 7(3), 64. https://doi.org/10.3390/jfmk7030064
  23. Drenowatz, C., Ferrari, G., Greier, K., Chen, S., & Hinterkörner, F. (2023). Physical fitness in Austrian elementary school children prior to and post-COVID-19. AIMS public health, 10(2), 480–495. https://doi.org/10.3934/publichealth.2023034
  24. Barnett, L. M., Hnatiuk, J. A., Salmon, J., & Hesketh, K. D. (2019). Modifiable factors which predict children's gross motor competence: a prospective cohort study. The international journal of behavioral nutrition and physical activity, 16(1), 129. https://doi.org/10.1186/s12966-019-0888-0
  25. Antunes, A. M., Freitas, D. L., Maia, J., Hedeker, D., Gouveia, É. R., Thomis, M., Lefevre, J., & Barnett, L. M. (2018). Motor performance, body fatness and environmental factors in preschool children. Journal of sports sciences,36(20), 2289–2295. https://doi.org/10.1080/02640414.2018.1449410

    Thanks again to you for your suggestions.

Reviewer 2 Report

dear authors , after reviewing the manuscript, please refer to my comments and improve your work in terms of:

Introduction:

Please elaborate in the introduction on the limitations imposed by the Pandemic. Part of the duration of the Pandemic was stationary in kindergartens or schools, if and what measures were taken to reduce the risk of contagion, and how these may have affected children's physical activity and motor skills. Please support your theses with relevant literature.

Material and methods:

Was the study anonymous? How was care taken to reduce errors in completing the questionnaire? Were the research participants trained and informed on how to complete the questionnaire?

Was the questionnaire on children's performance based only on the subjective judgement of the teacher? Were any objective measurements also taken?

In this section, please provide a diagram of the research experiment, including the number of participants at each stage, as well as the sampling criteria used.

-Were the inclusion and exclusion criteria applied and what were they?

Please describe with what criteria the intensity of the effort was determined, as I have not noticed this in any part of the article.

- In the supplementary materials, please attach the entire questionnaire used

Results

Please highlight the statistically significant results and present them in a graphical form, which will enable a more complete and clearer reading for the potential readers. 

Discussion

Please elaborate on the paragraph concerning the role of the teacher, his/her level of competence and the level of children's activity and motor skills, referring to relevant literature.

Limitations

Please indicate whether there were any limitations to the study and what was done to reduce or limit them.

Author Response

Thank you for the careful reading and constructive suggestions, the following are the revisions and responses to the relevant questions.

  1. The literature has been added and the description of the measures taken for the pandemic has been added.
  2. Added anonymous description in 2.1 Sociodemographic Variables

All study participants were given information about the link to read more details about the study and accepted online to participate in the study anonymously.

  1. This research is mainly purpose at the questionnaire survey conducted by Taiwan’s preschool educators. Due to the content of the questionnaire is to understand the curriculum implementation and children’s activities before and after the pandemic, preschool educators who have taught for less than three years must be excluded.
  2. This study did not perform any experimental operations, only collects questionnaires once. Each participant subjectively completes questionnaire on basic information, teaching of physical movement courses, performance of fundamental motor skills, how it was implemented during the pandemic (from January 2020 to November 2022), and using the same questions asked retrospectively, remembering the implementation before the pandemic (before January 2020).
  3. Added related figures to the article.
  4. The discussion had been rewritten, adding more relevant literature.
  5. Add limitations
  6. Add the following articles.
  1. de Figueiredo, C. S., Sandre, P. C., Portugal, L. C. L., Mázala-de-Oliveira, T., da Silva Chagas, L., Raony, Í., Ferreira, E. S., Giestal-de-Araujo, E., Dos Santos, A. A., & Bomfim, P. O. (2021). COVID-19 pandemic impact on children and adolescents' mental health: Biological, environmental, and social factors. Progress in neuro-psychopharmacology & biological psychiatry, 106, 110171. https://doi.org/10.1016/j.pnpbp.2020.110171
  2. Hosseinzadeh, P., Zareipour, M., Baljani, E., & Moradali, M. R. (2022). Social Consequences of the COVID-19 Pandemic. A Systematic Review. Investigacion y educacion en enfermeria, 40(1), e10. https://doi.org/10.17533/udea.iee.v40n1e10
  3. Matsubayashi, T., Ishikawa, Y., & Ueda, M. (2022). Economic crisis and mental health during the COVID-19 pandemic in Japan. Journal of affective disorders, 306, 28–31. https://doi.org/10.1016/j.jad.2022.03.037
  4. Brooks, S. K., Webster, R. K., Smith, L. E., Woodland, L., Wessely, S., Greenberg, N., & Rubin, G. J. (2020). The psychological impact of quarantine and how to reduce it: rapid review of the evidence. Lancet (London, England), 395(10227), 912–920. https://doi.org/10.1016/S0140-6736(20)30460-8
  5. Samji, H., Wu, J., Ladak, A., Vossen, C., Stewart, E., Dove, N., Long, D., & Snell, G. (2022). Review: Mental health impacts of the COVID-19 pandemic on children and youth - a systematic review. Child and adolescent mental health, 27(2), 173–189. https://doi.org/10.1111/camh.12501
  6. Meherali, S., Punjani, N., Louie-Poon, S., Abdul Rahim, K., Das, J. K., Salam, R. A., & Lassi, Z. S. (2021). Mental Health of Children and Adolescents Amidst COVID-19 and Past Pandemics: A Rapid Systematic Review. International journal of environmental research and public health, 18(7), 3432. https://doi.org/10.3390/ijerph18073432
  7. Tang, S., Xiang, M., Cheung, T., & Xiang, Y. T. (2021). Mental health and its correlates among children and adolescents during COVID-19 school closure: The importance of parent-child discussion. Journal of affective disorders, 279, 353–360. https://doi.org/10.1016/j.jad.2020.10.016
  8. Li, B., Ng, K., Tong, X., Zhou, X., Ye, J., & Yu, J. J. (2023). Physical activity and mental health in children and youth during COVID-19: a systematic review and meta-analysis. Child and adolescent psychiatry and mental health, 17(1), 92. https://doi.org/10.1186/s13034-023-00629-4
  9. Park, A. H., Zhong, S., Yang, H., Jeong, J., & Lee, C. (2022). Impact of COVID-19 on physical activity: A rapid review. Journal of global health, 12, 05003. https://doi.org/10.7189/jogh.12.05003
  10. Wunsch, K., Kienberger, K., & Niessner, C. (2022). Changes in Physical Activity Patterns Due to the Covid-19 Pandemic: A Systematic Review and Meta-Analysis. International journal of environmental research and public health, 19(4), 2250. https://doi.org/10.3390/ijerph19042250
  11. Tulchin-Francis, K., Stevens, W., Jr, Gu, X., Zhang, T., Roberts, H., Keller, J., Dempsey, D., Borchard, J., Jeans, K., & VanPelt, J. (2021). The impact of the coronavirus disease 2019 pandemic on physical activity in U.S. children. Journal of sport and health science, 10(3), 323–332. https://doi.org/10.1016/j.jshs.2021.02.005
  12. DeBoer M. D. (2019). Assessing and Managing the Metabolic Syndrome in Children and Adolescents. Nutrients, 11(8), 1788. https://doi.org/10.3390/nu11081788
  13. Konuthula, D., Tan, M. M., & Burnet, D. L. (2023). Challenges and Opportunities in Diagnosis and Management of Cardiometabolic Risk in Adolescents. Current diabetes reports, 23(8), 185–193. https://doi.org/10.1007/s11892-023-01513-3
  14. Julian, V., Ciba, I., Olsson, R., Dahlbom, M., Furthner, D., Gomahr, J., Maruszczak, K., Morwald, K., Pixner, T., Schneider, A., Pereira, B., Duclos, M., Weghuber, D., Thivel, D., Bergsten, P., & Forslund, A. (2021). Association between Metabolic Syndrome Diagnosis and the Physical Activity-Sedentary Profile of Adolescents with Obesity: A Complementary Analysis of the Beta-JUDO Study. Nutrients, 14(1), 60. https://doi.org/10.3390/nu14010060
  15. Schwarzfischer, P., Gruszfeld, D., Stolarczyk, A., Ferre, N., Escribano, J., Rousseaux, D., Moretti, M., Mariani, B., Verduci, E., Koletzko, B., & Grote, V. (2019). Physical Activity and Sedentary Behavior From 6 to 11 Years. Pediatrics, 143(1), e20180994. https://doi.org/10.1542/peds.2018-0994
  16. Krishnaratne, S., Littlecott, H., Sell, K., Burns, J., Rabe, J. E., Stratil, J. M., Litwin, T., Kreutz, C., Coenen, M., Geffert, K., Boger, A. H., Movsisyan, A., Kratzer, S., Klinger, C., Wabnitz, K., Strahwald, B., Verboom, B., Rehfuess, E., Biallas, R. L., Jung-Sievers, C., … Pfadenhauer, L. M. (2022). Measures implemented in the school setting to contain the COVID-19 pandemic. The Cochrane database of systematic reviews, 1(1), CD015029. https://doi.org/10.1002/14651858.CD015029
  17. BrzÄ™k, A.; Strauss, M.; Sanchis-Gomar, F.; & Leischik, R. (2021). Physical Activity, Screen Time, Sedentary and Sleeping Habits of Polish Preschoolers during the COVID-19 Pandemic and WHO’s Recommendations: An Observational Cohort Study. International Journal of Environmental Research and Public Health, 18,
  18. Chen, X., Hong, X., Gao, W., Luo, S., Cai, J., Liu, G., & Huang, Y. (2022). Causal relationship between physical activity, leisure sedentary behaviors and COVID-19 risk: a Mendelian randomization study. Journal of translational medicine, 20(1), 216. https://doi.org/10.1186/s12967-022-03407-6
  19. Bozzola, E., Barni, S., Ficari, A., & Villani, A. (2023). Physical Activity in the COVID-19 Era and Its Impact on Adolescents' Well-Being. International journal of environmental research and public health, 20(4), 3275. https://doi.org/10.3390/ijerph20043275
  20. Teich, P., Fühner, T., Bähr, F., Puta, C., Granacher, U., & Kliegl, R. (2023). Covid Pandemic Effects on the Physical Fitness of Primary School Children: Results of the German EMOTIKON Project. Sports medicine - open, 9(1), 77. https://doi.org/10.1186/s40798-023-00624-1
  21. Carcamo-Oyarzun, J., Salvo-Garrido, S., & Estevan, I. (2023). Actual and Perceived Motor Competence in Chilean Schoolchildren before and after COVID-19 Lockdowns: A Cohort Comparison. Behavioral sciences (Basel, Switzerland), 13(4), 306. https://doi.org/10.3390/bs13040306
  22. Carballo-Fazanes, A., Rodrigues, L. P., Silva, R., Lopes, V. P., & Abelairas-Gómez, C. (2022). The Developmental Trajectory of Motor Competence of Children That Lived the COVID-19 Confinement Period: A Four-Year Follow-Up Study in Portuguese Children. Journal of functional morphology and kinesiology, 7(3), 64. https://doi.org/10.3390/jfmk7030064
  23. Drenowatz, C., Ferrari, G., Greier, K., Chen, S., & Hinterkörner, F. (2023). Physical fitness in Austrian elementary school children prior to and post-COVID-19. AIMS public health, 10(2), 480–495. https://doi.org/10.3934/publichealth.2023034
  24. Barnett, L. M., Hnatiuk, J. A., Salmon, J., & Hesketh, K. D. (2019). Modifiable factors which predict children's gross motor competence: a prospective cohort study. The international journal of behavioral nutrition and physical activity, 16(1), 129. https://doi.org/10.1186/s12966-019-0888-0
  25. Antunes, A. M., Freitas, D. L., Maia, J., Hedeker, D., Gouveia, É. R., Thomis, M., Lefevre, J., & Barnett, L. M. (2018). Motor performance, body fatness and environmental factors in preschool children. Journal of sports sciences, 36(20), 2289–2295. https://doi.org/10.1080/02640414.2018.1449410

Thanks again to you for your suggestions.

Reviewer 3 Report

Dear editor, dear authors, thank you for the opportunity to carry out a review for International Journal of Environmental Research and Public Health. There are no any conflicts of interest of my own that might be perceived to have influenced my review.

 The authors of the manuscript entitled "Impact of COVID-19 Pandemic on Children’s Fundamental Motor Skills: A Study for the Taiwanese Preschoolers Teachers" explore the impact of teachers on children's FMS and physical activity performance during the pandemic, and tried to understand whether preschool teachers' physical activity teaching was affected by the pandemic.

 In my opinion, the study design and concept are useful. The aim of the study was clearly defined. I have a few comments and suggestions regarding to the methods and results section.  

Questionnaire - describe in more detail the methodology of evaluating the questionnaire and individual items. Did a 5-point Likert scale use throughout the questionnaire? What did 0 represent and what did 5 represent? Did you also find out the real time of movement? Real performance of children? Or was it a subjective view of preschool teachers? How did you rate the frequency? If you have such data, you could also add characteristics of the interviewees (e.g. age, gender, place of residence, length of experience, etc.).

 Results - add references to tables 2-4 in the text.

 Discussion - a more extensive confrontation of own results with the results of other authors is missing. Complete the limiting factors of the study.

 Line 47 - May 19 - which year? I'm guessing 2020. Fill in the year because it seems confusing.

 There are stylistic errors throughout the text, the manuscript needs to be carefully checked.

Finally, I recommend giving the authors a chance to revise the manuscript after minor revision.

Author Response

  1. Modified sections of the questionnaire, add description about Likert scale

2.Add limitations

  1. Line 47

Thanks to the reviewer for the comments, Taiwan’s control were implemented very well, and the pandemic did not break out until a year later, as annotated in the article on May 19, 2021

4.The discussion had been rewritten, adding more relevant literature.

5.Add the following articles.

  1. de Figueiredo, C. S., Sandre, P. C., Portugal, L. C. L., Mázala-de-Oliveira, T., da Silva Chagas, L., Raony, Í., Ferreira, E. S., Giestal-de-Araujo, E., Dos Santos, A. A., & Bomfim, P. O. (2021). COVID-19 pandemic impact on children and adolescents' mental health: Biological, environmental, and social factors. Progress in neuro-psychopharmacology & biological psychiatry, 106, 110171. https://doi.org/10.1016/j.pnpbp.2020.110171
  2. Hosseinzadeh, P., Zareipour, M., Baljani, E., & Moradali, M. R. (2022). Social Consequences of the COVID-19 Pandemic. A Systematic Review. Investigacion y educacion en enfermeria, 40(1), e10. https://doi.org/10.17533/udea.iee.v40n1e10
  3. Matsubayashi, T., Ishikawa, Y., & Ueda, M. (2022). Economic crisis and mental health during the COVID-19 pandemic in Japan. Journal of affective disorders, 306, 28–31. https://doi.org/10.1016/j.jad.2022.03.037
  4. Brooks, S. K., Webster, R. K., Smith, L. E., Woodland, L., Wessely, S., Greenberg, N., & Rubin, G. J. (2020). The psychological impact of quarantine and how to reduce it: rapid review of the evidence. Lancet (London, England), 395(10227), 912–920. https://doi.org/10.1016/S0140-6736(20)30460-8
  5. Samji, H., Wu, J., Ladak, A., Vossen, C., Stewart, E., Dove, N., Long, D., & Snell, G. (2022). Review: Mental health impacts of the COVID-19 pandemic on children and youth - a systematic review. Child and adolescent mental health, 27(2), 173–189. https://doi.org/10.1111/camh.12501
  6. Meherali, S., Punjani, N., Louie-Poon, S., Abdul Rahim, K., Das, J. K., Salam, R. A., & Lassi, Z. S. (2021). Mental Health of Children and Adolescents Amidst COVID-19 and Past Pandemics: A Rapid Systematic Review. International journal of environmental research and public health, 18(7), 3432. https://doi.org/10.3390/ijerph18073432
  7. Tang, S., Xiang, M., Cheung, T., & Xiang, Y. T. (2021). Mental health and its correlates among children and adolescents during COVID-19 school closure: The importance of parent-child discussion. Journal of affective disorders, 279, 353–360. https://doi.org/10.1016/j.jad.2020.10.016
  8. Li, B., Ng, K., Tong, X., Zhou, X., Ye, J., & Yu, J. J. (2023). Physical activity and mental health in children and youth during COVID-19: a systematic review and meta-analysis. Child and adolescent psychiatry and mental health, 17(1), 92. https://doi.org/10.1186/s13034-023-00629-4
  9. Park, A. H., Zhong, S., Yang, H., Jeong, J., & Lee, C. (2022). Impact of COVID-19 on physical activity: A rapid review. Journal of global health, 12, 05003. https://doi.org/10.7189/jogh.12.05003
  10. Wunsch, K., Kienberger, K., & Niessner, C. (2022). Changes in Physical Activity Patterns Due to the Covid-19 Pandemic: A Systematic Review and Meta-Analysis. International journal of environmental research and public health, 19(4), 2250. https://doi.org/10.3390/ijerph19042250
  11. Tulchin-Francis, K., Stevens, W., Jr, Gu, X., Zhang, T., Roberts, H., Keller, J., Dempsey, D., Borchard, J., Jeans, K., & VanPelt, J. (2021). The impact of the coronavirus disease 2019 pandemic on physical activity in U.S. children. Journal of sport and health science, 10(3), 323–332. https://doi.org/10.1016/j.jshs.2021.02.005
  12. DeBoer M. D. (2019). Assessing and Managing the Metabolic Syndrome in Children and Adolescents. Nutrients, 11(8), 1788. https://doi.org/10.3390/nu11081788
  13. Konuthula, D., Tan, M. M., & Burnet, D. L. (2023). Challenges and Opportunities in Diagnosis and Management of Cardiometabolic Risk in Adolescents. Current diabetes reports, 23(8), 185–193. https://doi.org/10.1007/s11892-023-01513-3
  14. Julian, V., Ciba, I., Olsson, R., Dahlbom, M., Furthner, D., Gomahr, J., Maruszczak, K., Morwald, K., Pixner, T., Schneider, A., Pereira, B., Duclos, M., Weghuber, D., Thivel, D., Bergsten, P., & Forslund, A. (2021). Association between Metabolic Syndrome Diagnosis and the Physical Activity-Sedentary Profile of Adolescents with Obesity: A Complementary Analysis of the Beta-JUDO Study. Nutrients, 14(1), 60. https://doi.org/10.3390/nu14010060
  15. Schwarzfischer, P., Gruszfeld, D., Stolarczyk, A., Ferre, N., Escribano, J., Rousseaux, D., Moretti, M., Mariani, B., Verduci, E., Koletzko, B., & Grote, V. (2019). Physical Activity and Sedentary Behavior From 6 to 11 Years. Pediatrics, 143(1), e20180994. https://doi.org/10.1542/peds.2018-0994
  16. Krishnaratne, S., Littlecott, H., Sell, K., Burns, J., Rabe, J. E., Stratil, J. M., Litwin, T., Kreutz, C., Coenen, M., Geffert, K., Boger, A. H., Movsisyan, A., Kratzer, S., Klinger, C., Wabnitz, K., Strahwald, B., Verboom, B., Rehfuess, E., Biallas, R. L., Jung-Sievers, C., … Pfadenhauer, L. M. (2022). Measures implemented in the school setting to contain the COVID-19 pandemic. The Cochrane database of systematic reviews, 1(1), CD015029. https://doi.org/10.1002/14651858.CD015029
  17. BrzÄ™k, A.; Strauss, M.; Sanchis-Gomar, F.; & Leischik, R. (2021). Physical Activity, Screen Time, Sedentary and Sleeping Habits of Polish Preschoolers during the COVID-19 Pandemic and WHO’s Recommendations: An Observational Cohort Study. International Journal of Environmental Research and Public Health, 18,
  18. Chen, X., Hong, X., Gao, W., Luo, S., Cai, J., Liu, G., & Huang, Y. (2022). Causal relationship between physical activity, leisure sedentary behaviors and COVID-19 risk: a Mendelian randomization study. Journal of translational medicine, 20(1), 216. https://doi.org/10.1186/s12967-022-03407-6
  19. Bozzola, E., Barni, S., Ficari, A., & Villani, A. (2023). Physical Activity in the COVID-19 Era and Its Impact on Adolescents' Well-Being. International journal of environmental research and public health, 20(4), 3275. https://doi.org/10.3390/ijerph20043275
  20. Teich, P., Fühner, T., Bähr, F., Puta, C., Granacher, U., & Kliegl, R. (2023). Covid Pandemic Effects on the Physical Fitness of Primary School Children: Results of the German EMOTIKON Project. Sports medicine - open, 9(1), 77. https://doi.org/10.1186/s40798-023-00624-1
  21. Carcamo-Oyarzun, J., Salvo-Garrido, S., & Estevan, I. (2023). Actual and Perceived Motor Competence in Chilean Schoolchildren before and after COVID-19 Lockdowns: A Cohort Comparison. Behavioral sciences (Basel, Switzerland), 13(4), 306. https://doi.org/10.3390/bs13040306
  22. Carballo-Fazanes, A., Rodrigues, L. P., Silva, R., Lopes, V. P., & Abelairas-Gómez, C. (2022). The Developmental Trajectory of Motor Competence of Children That Lived the COVID-19 Confinement Period: A Four-Year Follow-Up Study in Portuguese Children. Journal of functional morphology and kinesiology, 7(3), 64. https://doi.org/10.3390/jfmk7030064
  23. Drenowatz, C., Ferrari, G., Greier, K., Chen, S., & Hinterkörner, F. (2023). Physical fitness in Austrian elementary school children prior to and post-COVID-19. AIMS public health, 10(2), 480–495. https://doi.org/10.3934/publichealth.2023034
  24. Barnett, L. M., Hnatiuk, J. A., Salmon, J., & Hesketh, K. D. (2019). Modifiable factors which predict children's gross motor competence: a prospective cohort study. The international journal of behavioral nutrition and physical activity, 16(1), 129. https://doi.org/10.1186/s12966-019-0888-0
  25. Antunes, A. M., Freitas, D. L., Maia, J., Hedeker, D., Gouveia, É. R., Thomis, M., Lefevre, J., & Barnett, L. M. (2018). Motor performance, body fatness and environmental factors in preschool children. Journal of sports sciences, 36(20), 2289–2295. https://doi.org/10.1080/02640414.2018.1449410

Thanks again to you for your suggestions.

Round 2

Reviewer 1 Report

Dear authors,

thank you for the revision.

I considered it sufficient.

Best regards.

Reviewer 2 Report

Dear authors, thank you for carefully considering my comments and suggestions, I recommend the manuscript for publication, best regards